# Neutrophil Extracellular Traps Promote the Development and Growth of Human Salivary Stones

**DOI:** 10.3390/cells9092139

**Published:** 2020-09-22

**Authors:** Mirco Schapher, Michael Koch, Daniela Weidner, Michael Scholz, Stefan Wirtz, Aparna Mahajan, Irmgard Herrmann, Jeeshan Singh, Jasmin Knopf, Moritz Leppkes, Christine Schauer, Anika Grüneboom, Christoph Alexiou, Georg Schett, Heinrich Iro, Luis E. Muñoz, Martin Herrmann

**Affiliations:** 1Friedrich-Alexander-University Erlangen-Nürnberg (FAU), Department of Otolaryngology, Head and Neck Surgery, Universitätsklinikum Erlangen, Waldstrasse 1, 91054 Erlangen, Germany; mirco.schapher@uk-erlangen.de (M.S.); Michael.Koch@uk-erlangen.de (M.K.); Christoph.Alexiou@uk-erlangen.de (C.A.); Heinrich.Iro@uk-erlangen.de (H.I.); 2Friedrich-Alexander-University Erlangen-Nürnberg (FAU), Department of Internal Medicine 3, Universitätsklinikum Erlangen, Ulmenweg 18, 91054 Erlangen, Germany; Daniela.Weidner@uk-erlangen.de (D.W.); Aparna.Mahajan@uk-erlangen.de (A.M.); Irmgard.Herrmann@uk-erlangen.de (I.H.); Jeeshan.Singh@uk-erlangen.de (J.S.); Jasmin.Knopf@uk-erlangen.de (J.K.); Christine.Schauer@uk-erlangen.de (C.S.); Anika.Grueneboom@uk-erlangen.de (A.G.); Georg.Schett@uk-erlangen.de (G.S.); Luis.Munoz@uk-erlangen.de (L.E.M.); 3Friedrich-Alexander-University Erlangen-Nürnberg (FAU), Deutsches Zentrum für Immuntherapie, Universitätsklinikum Erlangen, Ulmenweg 18, 91054 Erlangen, Germany; Stefan.Wirtz@uk-erlangen.de (S.W.); Moritz.Leppkes@uk-erlangen.de (M.L.); 4Friedrich-Alexander-University Erlangen-Nürnberg (FAU), Institute of Functional and Clinical Anatomy, Universitätsstrasse 19, 91054 Erlangen, Germany; michael.scholz@fau.de; 5Friedrich-Alexander-University Erlangen-Nürnberg (FAU), Department of Internal Medicine 1, Universitätsklinikum Erlangen, Ulmenweg 18, 91054 Erlangen, Germany

**Keywords:** sialolithiasis, salivary stones, lithogenesis, stone development, stone growth, sialadenitis, salivary glands, neutrophils, neutrophil extracellular traps

## Abstract

Salivary gland stones, or sialoliths, are the most common cause of the obstruction of salivary glands. The mechanism behind the formation of sialoliths has been elusive. Symptomatic sialolithiasis has a prevalence of 0.45% in the general population, is characterized by recurrent painful periprandial swelling of the affected gland, and often results in sialadenitis with the need for surgical intervention. Here, we show by the use of immunohistochemistry, immunofluorescence, computed tomography (CT) scans and reconstructions, special dye techniques, bacterial genotyping, and enzyme activity analyses that neutrophil extracellular traps (NETs) initiate the formation and growth of sialoliths in humans. The deposition of neutrophil granulocyte extracellular DNA around small crystals results in the dense aggregation of the latter, and the subsequent mineralization creates alternating layers of dense mineral, which are predominantly calcium salt deposits and DNA. The further agglomeration and appositional growth of these structures promotes the development of macroscopic sialoliths that finally occlude the efferent ducts of the salivary glands, causing clinical symptoms and salivary gland dysfunction. These findings provide an entirely novel insight into the mechanism of sialolithogenesis, in which an immune system-mediated response essentially participates in the physicochemical process of concrement formation and growth.

## 1. Introduction

Salivary gland stones, or sialoliths, are the most common cause for the obstruction of the salivary glands [1,2]. Symptomatic sialolithiasis has a prevalence of 0.45% in the general population and is characterized by recurrent painful periprandial swelling of the affected gland [3]. Sialolithiasis accounts for one-third of all salivary gland disorders [4] and typically occurs in middle-aged adults. In 80% and 20% of patients with sialolithiasis, the stones affect the excretory duct system of the submandibular and parotid glands, respectively [5]. One-quarter (25%) of the patients develop more than one stone. Postmortem investigations in humans detected small concrements even in asymptomatic patients, particularly in the submandibular glands [6,7]. These small concrements may provide a basis for the development of larger stones, which finally become clinically symptomatic. Sublingual and other small salivary glands rarely harbor symptomatic sialoliths [8,9]. Stones found within the submandibular glands are usually larger than those of the parotid gland. The shapes of the stones depend on the sites of their formation, ranging from elongated to more round or oval when having developed in the duct system or in the parenchyma, respectively [10]. Ultrasound imaging and sialendoscopy are preferentially employed to detect the concrements [11,12,13,14]. The latter often need to be removed by sialendoscopy-assisted interventions such as basket extractions, mechanical fragmentation, or transoral or endoscopic–transcutaneous surgical approaches [15,16,17,18,19,20].

The mechanism that leads to the formation of sialoliths has still been elusive. So far, various concepts have been discussed, ranging from altered ion concentrations in saliva [21] to the presence of bacteria or foreign bodies in the ducts [22] to micro-calcifications of cell debris ejected from salivary gland cells [23]. Saliva of parotid glands contains less calcium ions than those of submandibular glands, making the former less susceptible to the formation of concrements [24]. This observation is reflected by the reported differing prevalence of sialolithiasis for both glands [5,25]. Parotid sialoliths have an average size of 6.4 mm (range 1–31 mm) and contain more organic and less inorganic compounds than submandibular stones, which have an average size of 8.3 mm (range 1–35 mm) [4,5,26]. In addition to calcium ions, phospholipids and secretory glycoproteins contribute to sialolith formation [23,27,28,29]. At a certain size, the concrements may obstruct the ducts.

The secretory activity of the gland, which is predominantly controlled by the autonomous nervous system, seems to influence this process [30,31]. The administration of the β-adrenoreceptor agonist isoprenaline and calcium gluconate into rats induced sialoliths and sialadenitis [32], while parasympathetic stimulation facilitates saliva secretion. Along with reduced salivary flow rates, the presence of leucocytes in saliva was observed in patients with recurrent parotitis [33,34]. A leucocyte influx into the ducts may be further aggravated by several factors, including a microbial ascend [21,22,23,35]. Salivary stasis and ongoing inflammation may finally result in glandular dysfunction, atrophy, or sclerosis [23,36,37].

While such concepts appear sound, the reason for the growth of sialoliths is unknown [23]. Current models seeking to explain the formation of sialoliths diverge substantially [6,7,21,22,23,33,34]. Our study shows that neutrophil extracellular trap (NET) formation is an essential step for sialolith development. Neutrophils can externalize their chromatin decorated with granular proteins, which is a process usually referred to as the formation of NETs [38]. NET formation is a type of inflammatory response that can be triggered by various factors, among them the contact of neutrophils with crystals such as calcium salts [39], cholesterol [40], or urate [41], and also by pH variations [42], foreign bodies [43], and bacteria [44]. NETs tend to aggregate and form aggNETs [45,46,47,48,49] that incorporate particular matter, cellular debris, and viable immune cells [50]. In synopsis with former observations and our present results, we hypothesize that aggNETs serve as a “glue” that agglomerates salivary calcium crystals and proteins, thus driving the assembly of macroscopic sialoliths.

## 2. Materials and Methods

### 2.1. Patients and Clinical Interventions

All patients of our study cohort with symptomatic sialolithiasis (*n* = 37; 20 males, 17 females; mean age 47.3 ± 13.7 years) received a clinical examination, ultrasound imaging, and a subsequent sialendoscopy to ensure the diagnosis. Diagnostic and therapeutic interventions were carried out in a tertiary referral center specializing in salivary gland diseases (FAU Medical School, Department of Otolaryngology, Head and Neck Surgery, University of Erlangen-Nuremberg, Erlangen, Germany), obtaining stones from submandibular (*n* = 30) and parotid glands (*n* = 7).

Submandibular stones were removed by transoral surgery, which allowed obtaining a complete unfractionated sialolith. Parotid stones were removed either in total during an open surgical approach in which the stone was removed as a whole, or by sialendoscopically-assisted basket extraction (when the concrement was small enough to evacuate it through the duct). In cases of larger parotid concrements that could not be evacuated in total due to their size or that were not obtained during an open surgical approach, stone fragments were collected after sialendoscopically controlled intraductal pneumatic lithotripsy and subsequent basket extraction.

### 2.2. Chemicals and Biochemicals

If not stated otherwise, we obtained all chemicals and biochemicals from Sigma-Aldrich/Merck (Darmstadt, Germany).

### 2.3. Salivary Stones Storage and Processing

Sialoliths were stored at room temperature (for µCT, calcium staining) or dry at −20 °C (immunofluorescence) until analysis. Some stones were embedded in methacrylate and cut into 2 µm sections with a rotary microtome (RM2265, Leica, Wetzlar, Germany); others were fixed for 4–12 h with paraformaldehyde and decalcified in Teitel buffer (140 g EDTA free acid; 90 mL NH_3_ [30%] in 500 mL H_2_O; pH = 7.2; adjusted with NH_4_OH) for up to 4 weeks. The stones used for dye penetration assays were incubated with trypan blue (100 µg/mL), propidium iodide (10 µg/mL), or acridin orange (10 µg/mL) at ambient temperature for 10 min and subsequently ground with a diamond file. By mechanical grinding with ceramic or diamond tools, we obtained gravel of sialoliths, and material from the surface, the intermediate, and the center of the stones was received by selectively skimming off the layers.

### 2.4. Micro-Computed Tomography

For µCT scanning of the calculi, we used a cone-beam desktop microcomputer tomograph (µCT 40, Scanco Medical AG, Bruettisellen, Switzerland). The samples were positioned, fixed, and scanned (voxel size 9 µm, 45 kVp). µCT images were processed and analyzed quantitatively with Photoshop CS5 64 Bit (Adobe, Munich, Germany).

### 2.5. Digital Reconstruction of Volumetric µCT Data

Based on the volume data generated in CT, a physically based volume-rendering algorithm using a Monte Carlo path tracing method simulated the complex interactions of photons (emission, absorption, scattering) within the scanned specimens [51,52]. Cinematic Rendering generated photorealistic images by calculating realistic lighting by light transport simulation along hundreds or thousands of photons paths per pixel, using a stochastic process. Thus, even complex effects such as ambient occlusion or tissue density could be modeled.

### 2.6. Macrophotography

Macro images were taken with a Nikon 700 camera (Nikon, Tokyo, Japan) with a complementary metal oxide sensor (CMOS) in FX format (36.0 × 23.9 mm, 12.87 million pixels) and two different objectives (Nikon AF-S Nikkor 60 mm/2.8G ED; Nikon AF-S Nikkor 105 mm 1:2,8G VR). For illumination, we employed white light.

### 2.7. Microscopy

We used a BZ-X700 automated video microscope for fluorescence microscopy, equipped with Z-stack and stitching technology to increase the depth of field and the size (Keyence Corporation, Osaka, Japan). For bright field microscopy, oblique illumination was available. We used Photoshop CS5 or CC2018 (Adobe, Munich, Germany) for post-processing pictures and morphometry. Areas for morphometry were selected by a blinded investigator.

### 2.8. Optical Clearing and Light Sheet Fluorescence Microscopy (LSFM) of Submandibular Sialoliths

We fixed and decalcified submandibular sialoliths with paraformaldehyde and Teitel buffer, respectively, as described above. DNA of both decalcified and non-decalcified sialoliths was stained with propidium iodide (10 μg/mL) in PBS for 5 days at ambient temperature. We performed optical clearing according to established protocols [53]. Briefly, samples were dehydrated in ethanol series of 50%, 70%, and twice at 100% (*v/v*). Each dehydration step was carried out at room temperature for 2 days in gently shaken 5 mL tubes. After dehydration, samples were transferred to ethyl cinnamate, optically cleared for one day at ambient temperature, and imaged with a LaVison BioTec Ultramicroscope II including a LaVision BioTec Laser Module (LaVision BioTec GmbH, Bielefeld, Germany), an Olympus MVX10 zoom body (Olympus Germany, Hamburg, Germany), and an Andor Neo sCMOS Camera (Andor, Belfast, UK) with a pixel size of 6.5 μm. We used detection optics with fourfold optical magnification and NA 0.5. Autofluorescence was excited by a 488 nm optically pumped semiconductor laser (OPSL) and detected at 525/50 nm. For propidium iodide excitation, a 561 nm OPSL and a 620/60 nm emission filter were employed. For 3D reconstruction and optical clipping of LSFM data, we used Imaris software (Version 9.1, Bitplane AG, Zurich, Switzerland).

### 2.9. Von Kossa Staining for Mineralized Areas

We rinsed methacrylate sections several times with distilled water, incubated them with a 5% aqueous silver nitrate solution in the dark for 20 min, and finally exposed them to UV light until a black precipitate had formed. The remaining ionic silver was removed by incubation with 5% sodium thiosulfate for 5 min and extensive rinsing with distilled water.

### 2.10. DNA Staining

We stained the extracellular DNA of stones and histological samples with 10 µg/mL and 1 µg/mL propidium iodide in isotonic PBS (both from Thermo Fisher Scientific, Waltham, MA, USA) for at least 10 min at ambient temperature. Digestions with DNase 1 (1 U/mL; 10 mM Mg^2+^; 5 mM Ca^2+^) were performed for 120 min at 37 °C to confirm extracellular DNA as the origin of the observed propidium iodide signals.

### 2.11. Immunostaining for Neutrophil Markers

Histological sections and gravels from sialoliths were analyzed for extracellular DNA, neutrophil elastase, and citrullinated histone H3. Samples were fixed with 4% paraformaldehyde for 10 min and blocked for 18 h at 4 °C in PBS containing 10% FBS (Merck Millipore, Billerica, Waltham, MA, USA). Primary antibodies detecting neutrophil elastase or citrullinated histone H3 (Abcam, Cambridge, UK; ab21595 and ab1503, respectively) were used following manufacturer’s recommendations. AffiniPure Cy5-conjugated Goat-Anti-Rabbit IgG (H + L) secondary antibodies (Jackson Immuno Research Labs, West Grove, PA, USA) were co-incubated with 1 µg/mL Hoechst stain or propidium iodide. Slides were embedded in DAKO fluorescent mounting medium (Agilent Technologies, Santa Clara, CA, USA) according to manufacturer´s recommendations.

### 2.12. Neutrophil Elastase Activity in Human Sialoliths

First, 1 mg of sanding dust from sialoliths (diamond file, grain size 300) was resuspended in 500 µL of PBS. Then, 25 µL of this material (or H_2_O as controls) were added to 175 µL of PBS and 25 µL of 1 M fluorogenic substrate MeOSuc-AAPV-AMC (Santa Cruz Biotechnology, Heidelberg, Germany; sc-201163). Samples were continuously analyzed in black 96-well plates (Thermo Fisher Scientific, Waltham, MA, USA; 137101) every 10 min for up to 12 h at 37 °C in a thermostated fluorescence reader (TECAN Infinite 200 Pro, Switzerland; excitation 360 nm; emission 465 nm).

### 2.13. DNA Extraction, Sequencing, and Microbiome Analysis

To remove DNA contaminations, we washed the surfaces of the sialoliths extensively with at least 3 changes of >20-fold volumes of PBS. We ground the stones with a gentleMACS Dissociator (Miltenyi Biotec, Bergisch Gladbach, Germany) using M-tubes and TE buffer (100 mM Tris-Cl, 10 mM EDTA, pH = 8.0). After phase separation, we recovered the sialoliths-borne DNA by adding ammonium acetate to a final concentration of 2 M to the aqueous phase, followed by 0.6 volumes of isopropanol and overnight incubation at −20 °C. The pellet was redissolved in 300 mM sodium acetate and reprecipitated by the addition of 2 volumes of absolute ethanol (12 h at −20 °C). An aliquot of the DNA samples was run on an agarose gel.

The 4 samples with the highest DNA content were analyzed by 16S-based metagenomic sequencing as described elsewhere [54]. The V3–V4 region of the bacterial 16S rRNA gene was amplified by PCR (35 cycles; 98 °C for 15 s, 58 °C for 20 s, 72 °C for 40 s; NEB Next Ultra II Q5 Master Mix, New England Biolabs, Ipswich, MA, USA) using template DNA and region-specific primers (containing barcodes and Illumina flow cell adaptor sequences; Eurofins Genomics, Munich, Germany). The amplicons were purified with Agencourt AMPure XP Beads (Beckmann Coulter, Brea, CA, USA), normalized, and pooled. Sequencing was performed with an Illumina MiSeq device using a 600-cycle paired-end kit and the standard Illumina HP10 and HP11 sequencing primers. The read Fastq files were bioinformatically processed (merging, demultiplexing, quality filtering, dereplication, chimera removal) using the 64-bit version of Usearch 10 according to the Uparse pipeline [55]. Operational taxonomic units (OTUs) were selected at a threshold of 97% similarity and taxonomically classified by comparing the representative OTU sequence to the reference file of the ribosomal database project (RDP version 16).

### 2.14. Statistical Analysis

One-way ANOVA was used to calculate significances of the differences between independent groups with the SPSS Software (IBM, Armonk, NY, USA, version 24). *p*-values ≤ 0.05 were considered statistically significant.

### 2.15. Study Approval

A written informed consent was obtained from each patient concerning diagnostic procedures, treatment to remove salivary stones, and the subsequent use of these specimens for research purposes, including data analysis, prior to inclusion in the study. The study was approved by the University´s ethical review committee (186_19 Bc).

## 3. Results

### 3.1. Human Sialoliths are Macroscopically Polymorphic

Sialoliths show individual characteristics in size, shape, and color. Their surface structure ranges from smooth to fissured, underlining the pleomorphic nature of the stones. Many appear to be composites of smaller spheric building blocks (Figure 1). Detailed analyses of the mineral composition of sialoliths were described elsewhere [56,57].

### 3.2. Sialoliths Consist of an Onion Skin-Like Shell Structure of Extracellular DNA and Calcified Layers

Micro-computed tomography (µCT) showed concentric alternating radiodense (calcified) and translucent (organic) layers in submandibular and parotid sialoliths (Figure 2a–c). The 3D reconstruction of complete stones confirmed their layered structure, illustrating this alternating pattern that extended in all examined stones from the center to the surface (Figure 2d). The center of all examined stones was found to be radiolucent, indicating a lower degree of mineralization and a higher degree of organic compounds. Von-Kossa staining identified the radiodense layers as crystalline calcium deposits, which were surrounded again by more radiolucent, organic layers (Figure 2e).

Propidium iodide staining of hard-cut methacrylate embedded stones indicated that the radiolucent, organic regions contain extracellular DNA (Figure 2f, displayed in red). Merged images confirmed that these extracellular DNA containing layers alternated with the dense calcium crystal layers (Figure 2f, displayed in green). The extracellular DNA signals were predominantly detected as extended patches of aggregated chromatin.

### 3.3. The Calcium-Containing Layers of Sialoliths are Responsible for their Tightness

The incubation of whole native sialoliths in different staining solutions objectified the compactness of the calcified outer shells (acridine orange, MW = 265 g/mol; trypan blue, MW = 873 g/mol; propidium iodide, MW = 668 g/mol). A subsequent sanding of the stones revealed that parts of the interior were protected from the penetration of any employed dye (Figure 3a–f). These dye spared regions were found in every single stone, particularly in the center. Occasionally, small caves in the stones´ interior communicated with the surface via small clefts (Figure 3e). The surfaces of the inner caverns were also stained, but their walls were as impermeable for all employed dyes, as were the outer shells.

The detected propidium iodide signals in fluorescent microphotographs illustrated a diffuse distribution of extracellular DNA on the surface of native stones (Figure 3f, left panel). After decalcification, propidium iodide penetrated into the stones´ inner layers, and the staining was then detected throughout the entire sialolith (Figure 3f, right panel). The center of the sialolith displayed a strong autofluorescence that distinguished the regular shaped, homogenous core from the jagged and fissured outer layers. However, DNA was to be detected all over the sialolith, including the core (Figure 3f, Appendix A). This confirmed that the calcification was responsible for the dye impermeability of the stones´ outer layers. To verify that the propidium iodide signals in the stones´ interior were caused by extracellular DNA, we incubated sialolithic gravel with propidium iodide. The obtained intense staining signals were DNase 1 sensitive and considerably reduced after the digestion, identifying the origin of the signal as extracellular DNA (Figure 3g).

### 3.4. The Majority of Sialoliths Harbor Bacterial DNA

As bacteria [58] and calcium salt crystals are established inducers of NETs [43,47,49,59], we tested whether bacteria are to be found in the sialoliths. The DNA was extracted from gravel obtained from 28 submandibular concrements. A subsequent bacteria-specific 16S rRNA PCR showed that 23/28 (82.1%) of the sialoliths contained bacterial DNA. To identify the involved species, we genotyped the four preparations with the highest yields of bacterial DNA and established the microbiome of these stones. We detected various bacterial taxa typically present in the oral (Streptococcus dentisani [60]; Streptococcus intermedius [61]; Fusobacterium nucleatum [62]; Prevotella spp. [63]) and intestinal microflora (Barnesiella intestinihominis [64]; Porphyromonadaceae [65]). In addition, we observed the genus Phenylobacterium, which was reported to be present in the middle ear [66], putatively being a commensal of the upper aerodigestive tract. All these taxa were present in every stone we tested. Interestingly, none of these bacteria were detected in the microbiome of gallstone controls. These results suggest that oral bacteria, in addition to calcium salt crystals, are inducers of the NETs that drive sialolith formation.

### 3.5. NETs are Involved in Sialolithogenesis

Next, we asked whether the patchy propidium iodide signals originated from NETs and co-stained gravel specimens for extracellular DNA, neutrophil elastase, and citrullinated histone H3. Neutrophil elastase and citrullinated histone H3 (citH3) could be robustly detected in sialoliths (Figure 4a) and were shown to colocalize with the extracellular DNA. This finding strengthened the hypothesis that the DNA patches observed in the sialoliths originate from NETs. To confirm the presence of enzymatically active neutrophil elastase within the sialoliths, we selectively harvested gravel material from the stones´ centers, intermediate parts, and surfaces, which was subsequently ground and analyzed for neutrophil elastase activity. We detected a considerable enzyme activity in all stones and in all areas. A tendency to higher activities at the surfaces was noted, the latter representing the newest parts of the sialoliths (Figure 4b).

## 4. Discussion

Although the development of a clinical sialadenitis is not per se associated with the presence of sialoliths and vice versa, these two constellations occur in parallel, mutually reinforcing each other [23]. Both clinical pictures are associated with a disturbed gland function, a reduced salivary flow rate, and the presence of leucocytes in saliva [33,34]. Altered ion concentrations in saliva [21] leading to crystal formation as well as the presence of cellular debris, foreign bodies, or bacteria have been considered to contribute to sialolith formation in the past [22,23,24,67,68,69]. The most solid observations on the development of salivary stones describe the occurrence of sialomicroliths, having been found to be present in the salivary glands of asymptomatic individuals [6]. At a certain size, they may cause ductal congestions and initiate salivary stasis, resulting in further inflammation, swelling, and finally sclerosis or atrophy of the adjacent tissue. However, it was unknown to date whether all these different mechanisms are interconnected and how microscopic crystals in the salivary ducts can form and grow to macroscopic concrements.

Crystals and other danger signals (i.e., bacteria) are potent attractors for neutrophils, inducing their activation and NET formation [43,47,49,59]. In the salivary glands, neutrophils enter the excretory duct system, which is presumably intensified by the presence of ascending bacteria, foreign bodies, and other chemoattractive factors as complement components. By externalization of their chromatin, neutrophils form neutrophil extracellular traps (NETs). NETs were already shown to promote the development of depositions in other organs in vivo [40,46]. Sodium urate and calcium carbonate crystals reportedly induce the extrusion of neutrophil DNA [43,47,49,59]. As salivary stones consist of calcium-based crystals (hydroxylapatite, brushite, and whitlockite) [70], we hypothesize that NET formation fuels the process of sialolithogenesis.

We show that the surface of salivary stones displays robust neutrophil elastase activity and is covered by an abundance of extracellular DNA, suggesting neutrophil recruitment and NET formation. The high concentrations of bicarbonate ions present in saliva are known to be an important NET formation trigger [49], additionally explaining the plenty of extracellular DNA that covers sialoliths. The clinical experience whereby patients with chronic recurrent sialadenitis respond favorably to the anti-inflammatory action of glucocorticoids instilled into the affected ducts [33] further strengthens the role of neutrophil-based inflammation in sialolith formation.

The onion skin-like distribution of calcium depositions, associated extracellular DNA accumulations, neutrophil elastase activity, and presence of citrullinated histone H3 argue for a step-wise, episodic growth of sialoliths. The low abundance of citrullinated histone H3 in the center of sialoliths can be explained by the fact that the less basic citrullinated molecules are lost from chromatin over time [71]. More intense signals can be found in the outer layers, as these represent younger parts of the sialoliths. In this context, a desirable double immunflourescence staining of neutrophil elastase and citrullinated histone H3 is technically complicated. Four different antibodies must be separately titrated on the same sample. In order to detect specific signals, we performed these stains on separate samples of the same material (Figure 4a). Furthermore, we have not detected any intact bacteria in salivary stones. Due to the presence of NETs containing toxic histones and active proteases, the visualization of complete bacteria is not expected within the stones. Therefore, we chose to extract the bacterial DNA, which further allowed us to genotype and identify the different species more precisely.

By capturing particulate matter and preventing its spreading, NETs generally help to reduce harm to the organism. In this scenario, the same mechanism facilitates the development of sialoliths. Our data explain the agglomeration of the building blocks of sialoliths by aggNETs, which represent the common final path in sialolithogenesis. The development of the stones seems to be based on a repeated process of neutrophil attraction, presumably intensified by bacterial colonization, NET formation, and the assembly of calcium-containing depositions. The subsequent aggregation to larger concrements, packaging, stabilization, and further calcification of the layers result in the appositional growth of the salivary stone.

## 5. Conclusions

NETs promote calcium crystal aggregation in the salivary ducts and trigger concrement formation. Different previous models trying to explain the process of sialolithogenesis can be brought into line by the induction of an inflammatory reaction, causing an enhanced influx of neutrophils into the salivary ducts. The activation of these neutrophils and the formation of NETs, enhanced by the presence of crystals and other danger signals as bacteria, results in the gradual development and appositional growth of salivary stones, which finally congest the excretory ducts of the gland. This immune system-based mechanism in the development of concrements may also apply for other organs within the body. In addition to the existing endoscopic or surgical treatment options, targeting neutrophils and NET formation may thus become a valuable instrument to prevent the development of salivary stones.

## Figures and Tables

**Figure 1 cells-09-02139-f001:**
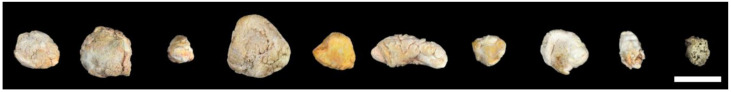
Sialoliths are unique in shape and surface composition. Ten representative submandibular stones (*n* = 30) under oblique white light, illustrating the individual surface differences between stones and even within the same sialolith. Note the different shapes and appearances that suggest a discontinuous and variable sialolithogenesis in differently shaped molds. Scale bar: 10 mm.

**Figure 2 cells-09-02139-f002:**
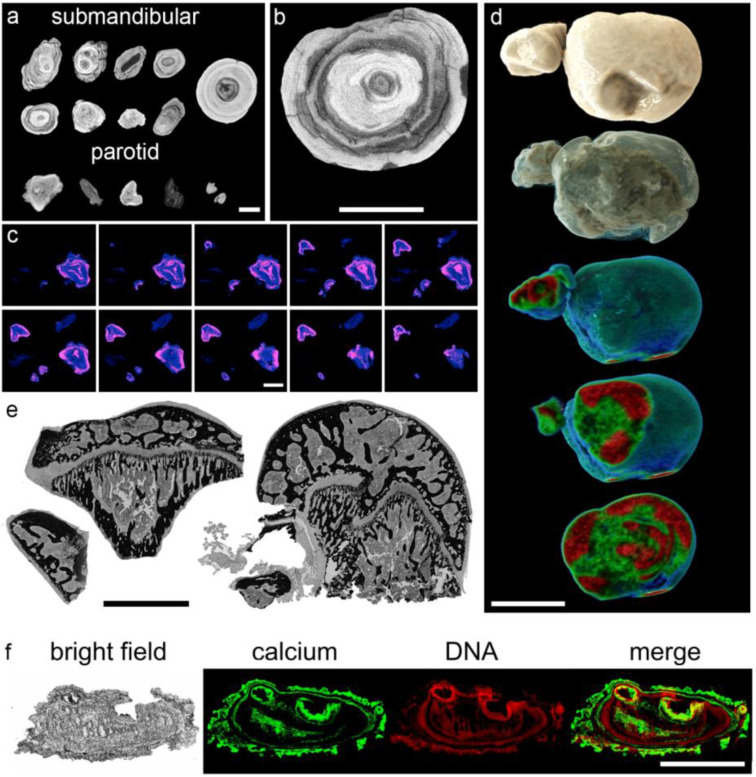
Sialoliths show onion skin-like layers, indicating a discontinuous formation process. (**a**) Micro-computed tomography (Micro-CT) images reveal concentric layers of calcifications (bright) around the center of submandibular (*n* = 9) and parotid sialoliths (*n* = 5), alternating with more radiotranslucent (dark) areas. (**b**) Magnification of a micro-CT image of a submandibular sialolith shown in (**a**). (**c**) Follow-up sections of artificially stained complete parotid sialoliths (*n* = 5; blue and pink reflect X-ray translucent and radiodense areas, respectively). (**d**) Representative 3D CT scan reconstructions of a submandibular sialolith (*n* = 5). Houndsfield units reflecting the radiodensity are represented by color codes (green reflects X-ray translucent and red reflects radiodense areas, respectively). (**e**) Representative Von-Kossa staining of methacrylate sections of submandibular sialoliths (*n* = 9) revealed that mineralized, predominantly calcium-containing areas (black) surround regions of organic depositions (gray). The observed onion skin-like layers are partially traversed by small septs, additionally enclosing organic compounds. (**f**) Calcifications (von-Kossa stain, negatively displayed in green) alternate with organic layers that harbor extracellular DNA (stained with propidium iodide, displayed in red), which are representatively shown in a submandibular sialolith (*n* = 11). Scale bar (**a**–**f**): 5 mm.

**Figure 3 cells-09-02139-f003:**
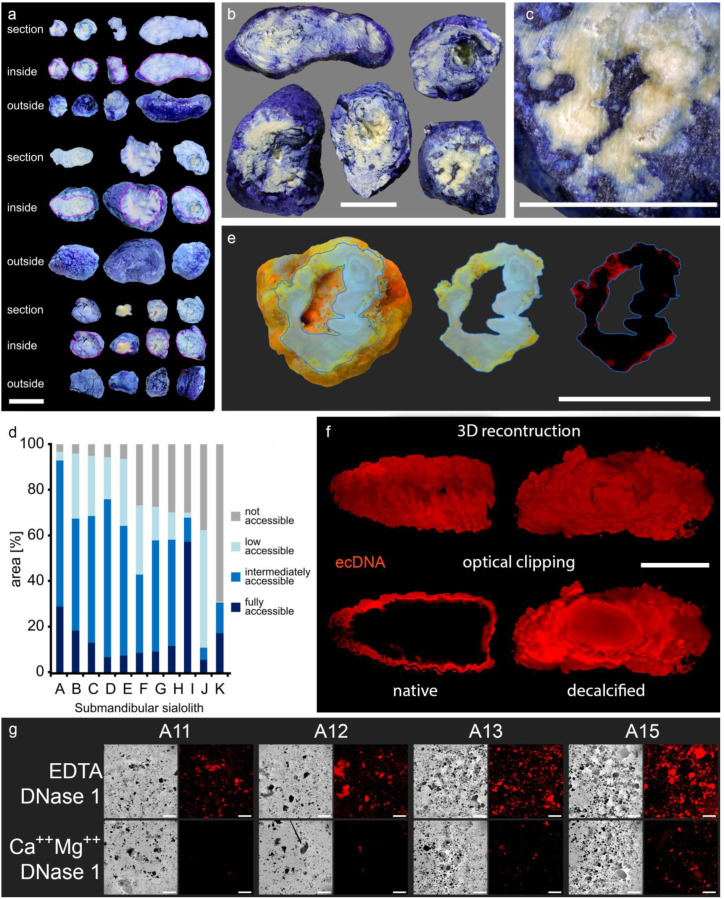
Sialoliths contain dye-impermeable, densely calcified areas and extracellular DNA (ecDNA). Stones were immersed in different dyes and subsequently ground to reveal their interior. The calcified surface is very compact and prevents the penetration of all employed dyes into the stones´ interior. (**a**–**c**) Trypan blue (MW = 873 g/mol) incompletely penetrates the native calculi, which is shown here in submandibular sialoliths (*n* = 11). (**d**) Morphometry of the submandibular sialoliths depicted in (**a**–**c**): every single stone shows dye inaccessible parts, particularly in the inner layers, quantified by image analysis (*n* = 11). (**e**) Acridin orange (MW = 265 g/mol) reveals that the surface of sialoliths occasionally communicates with cavities in the interior via small clefts (displayed in red, right panel; *n* = 5). (**f**) Representative LSFM (light sheet fluorescence microscopy) of a submandibular sialolith (*n* = 4). In native stones, propidium iodide (red, MW = 668 g/mol) only stains the extracellular DNA (ecDNA) at the surface. After decalcification, the dye penetrates the sialolith, demonstrating the role of the calcification for the compactness of the surface and indicates the containment of extracellular DNA within the entire stone, including the core. (**g**) Representative propidium iodide staining of stone gravel reveals the widespread distribution of extracellular DNA within the stones. The signals attenuated after treatment with DNase 1, confirming extracellular DNA as the signal´s origin (*n* = 7). Scale bar (**a**–**c**,**e**,**f**): 10 mm; scale bar (**g**): 200 µm.

**Figure 4 cells-09-02139-f004:**
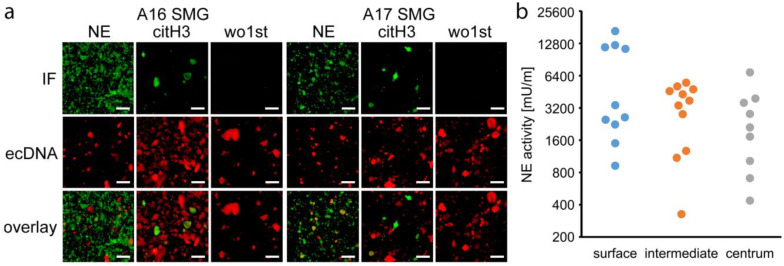
Sialoliths contain extracellular DNA, neutrophil elastase, and citrullinated histone H3. (**a**) Representative staining of neutrophil elastase (NE), citrullinated histone H3 (citH3), and extracellular DNA (ecDNA) in gravel of submandibular sialoliths (SMG: submandibular gland sialolith; IF: immunofluorescence; wo1st: without first antibody, negative control, *n* = 5). The co-localization of these molecules (overlay) indicate neutrophil extracellular traps (NETs) as building blocks of the sialoliths. (**b**) Neutrophil elastase activity was present in all parts of the sialoliths (quantified with a specific fluorogenic substrate, *n* = 11). The highest activities were detected in the superficial layers. Scale bar: 200 µm.

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
