# Peer review of "Neutrophil Extracellular Traps Promote the Development and Growth of Human Salivary Stones"

_cells, 2020, doi:10.3390/cells9092139_

Round 1

Reviewer 1 Report

Schapher et al. studied immunohistochemistry, immunofluorescence, CT scans and reconstruc-tions, special dye techniques, bacterial genotyping and enzyme activity analyses of sialoliths, indicating that neutrophil extracellular traps (NETs) initiate the formation and growth of sialoliths in humans. I have some comments. 1. The authors had better add the backgrounds of subjects.

Author Response

Reviewer 1:

"The authors had better add the backgrounds of subjects."

Answer:
Thank you for the comment. We revised the manuscript according to your recommendation and added demographic data of the subjects studied.

Reviewer 2 Report

Schapher et al. present an interesting study about development and growth of human salivary stones and the implication of NETs therein. 

The study is well designed. Minor things should be addressed before acceptance.

Line 66: What is an average size of concrements in submandibular and parotid glands, respectively?

Overall, please check font size (e.g. line 60, 71, 76, 100, 113).

The introduction, methods and discussion cover all important information And the findings are really nice! However, in the results section, it would help to work on the presentation of figure 2f and 4a. Especially with fig 4a, it was a bit difficult to understand at first what is meant with IF. Maybe it can be helpful to rearrange the images, e.g. have the first (upper) row of images with IF, since the according stain is just written above (NE, citH3, wo1st). Then the ecDNA, and finally the merge (lowest row). It might be easier for the reader to understand the figure. In the same sense, for fig 2f, the merge could be displayed on the right side, after the separate channels. 

Some questions, which do not need to be addressed experimentally, but I would like to maybe see in the discussion: For the visualisation of NETs, can a co-staining with citH3 and NE be achieved? Can bacteria be visualised within the calculi? 

Author Response

Reviewer 2:

Comment 2.1. Line 66: What is an average size of concrements in submandibular and parotid glands, respectively?

Answer:
Thank you for the comment. We revised the manuscript according to your recommendation and added the average sizes of the salivary stones (see lines 66-68).

Comment 2.2. Overall, please check font size (e.g. line 60, 71, 76, 100, 113).

Answer:
Thank you for your observation. We have used the template from MDPI and checked that the font size all through the source file is correct.  

Comment 2.3. The introduction, methods and discussion cover all important information And the findings are really nice! However, in the results section, it would help to work on the presentation of figure 2f and 4a. Especially with fig 4a, it was a bit difficult to understand at first what is meant with IF. Maybe it can be helpful to rearrange the images, e.g. have the first (upper) row of images with IF, since the according stain is just written above (NE, citH3, wo1st). Then the ecDNA, and finally the merge (lowest row). It might be easier for the reader to understand the figure. In the same sense, for fig 2f, the merge could be displayed on the right side, after the separate channels. 

Answer:

We have rearranged the figures 2f and 4a as suggested by the reviewer and explained all abbreviations in the figure legend.

Comment 2.4. Some questions, which do not need to be addressed experimentally, but I would like to maybe see in the discussion: For the visualisation of NETs, can a co-staining with citH3 and NE be achieved?

Answer:
Thank you for the comment. A double immunflourescence staining is technically complicated since 4 antibodies must be separately titrated on the same tissue to be sure that the staining is specific. This is impossible to achieve with salivary stones.

Comment 2.5. Can bacteria be visualised within the calculi? 

Answer:
Thank you for the comment. We have not visualized any intact bacteria in the salivary stones. Due to the presence of NETs containing toxic histones and active proteases within the stones, the visualization of complete bacteria is not expected. We therefore chose to extract the bacterial DNA out of the stones, which further allowed us to genotype and identify the different species more precisely.

Reviewer 3 Report

This manuscript provides an insight into the role of neutrophil extracellular traps in sialolithogenesis. This is a well-planned study. manuscript is organized and clearly written. Methods are adequately described and results are clearly presented and easy to understand. Conclusions are supported by results. However, the discussion section is a little weak and has scope for improvement. Results/ are different experiments are not discussed wrt each other. Discussing how results/observations from different experiments compliment/support each other will help in highlighting the importance of these results.

Author Response

Reviewer 3:

Comment 3.1. This manuscript provides an insight into the role of neutrophil extracellular traps in sialolithogenesis. This is a well-planned study. manuscript is organized and clearly written. Methods are adequately described and results are clearly presented and easy to understand. Conclusions are supported by results. However, the discussion section is a little weak and has scope for improvement. Results/ are different experiments are not discussed wrt each other. Discussing how results/observations from different experiments compliment/support each other will help in highlighting the importance of these results.

Answer:
Thank you for your comment. We revised the manuscript thoroughly, edited the discussion section and added further aspects to support and discuss the results.

Reviewer 4 Report

i don't find nothing about ethical approval of the study

patients gived informed consent?

which kind of speciific addtional methods were employed to justify the final result showing that stones are from endogenous neutrophil production and not "contamination" from other sources?

Author Response

Reviewer 4:

Comment 4.1. i don't find nothing about ethical approval of the study; patients gived informed consent?

Answer:
Thank you for the comment. The ethical approval and information about the informed consent is described in the materials and methods section (please see lines 205-208).

Comment 4.2. which kind of speciific addtional methods were employed to justify the final result showing that stones are from endogenous neutrophil production and not "contamination" from other sources?

Answer:
Thank you for the comment. After endoscopic or surgical removal of the stones, a contamination with "foreign" neutrophils (e.g. from the oral cavity of the patient) is just possible on the stone´s surface. However, NETs and NE activity can be found throughout the stones, also in the interior, as depicted in figure 4b. These neutrophils are included within the stones and cannot be a contamination from the outside.

Round 2

Reviewer 1 Report

This manuscript is acceptable for publication.

Reviewer 4 Report

I agree with the revision of the manuscript

for me it can accepted for publication